# Research on the Comprehensive Evaluation Method of Driving Behavior of Mining Truck Drivers in an Open-Pit Mine

**Zhao Zhang** [1,*] , **Ruixin Zhang** [1,2] **and Jiandong Sun** [2,3]

1    School of Energy and Mining Engineering, China University of Mining and Technology (Beijing), Beijing 100083, China; zhangrx139@163.com
2    Mine Safety Institute, North China Institute of Science and Technology, Langfang 065201, China; sjd_xx@126.com
3    State Key Laboratory of Coal Resources and Safe Mining, China University of Mining and Technology, Beijing 100083, China
*    Correspondence: zhangzhao821@126.com; Tel.: +86-188-1062-0637

**Abstract:** Trucking is an important production link in most open-pit mines, and its transportation cost accounts for more than 50% of the total production cost of open-pit mines. The quality of the driver's driving behavior plays a crucial role in the fine control of the production cost of transportation. Different from the previous evaluation studies of drivers' driving behavior in open-pit mines, which mainly took safety driving behavior index as a factor variable, this paper puts forward a comprehensive evaluation method of driving behavior of mining truck drivers, which takes both safety driving and transportation cost as factor variables. Taking the mining truck as the research object, firstly, a scientific and reasonable data collection scheme is established, and the data information characterizing the transport state of the mining truck is obtained through data collection and analysis. Secondly, the RKNN algorithm of time series prediction and the wavelet analysis method are used to achieve noise reduction and missing processing of the original data so as to obtain accurate sample data. Then, taking the principal component analysis method as the entry point, through constructing the principal component analysis theory model, the key index system representing safe driving behavior and transportation cost is established to realize the comprehensive evaluation of the driving behavior of mining truck drivers, and the evaluation system of "standard driving", "prudent driving" and "aggressive driving" of mining truck drivers is formulated. The results show that after noise reduction, the accuracy of mining car operation data can be improved by 7~12%, and the transportation cost can be reduced by about 5% after the driver's operation behavior is standardized.

**Keywords:** open-pit mine; mining truck; driving behavior evaluation; principal component analysis

## 1. Introduction

Currently, as a subversive innovative technology, intelligence has become the core driving force for the revolution of basic industries worldwide [1]. By adopting intelligent and new technology, developing and utilizing mining information resources, optimizing the whole process of mining activities, and improving the management level and technology level of the mining industry, the Chinese mining industry can be promoted to the long-term goal of safety, high efficiency, economy, green, and sustainable development [2,3]. In the process of open-pit mining, the transportation link is an essential production link in the process of open-pit mining, and truck transportation is the main mode of transportation in most domestic open-pit mines [4]. In the production cost of an open-pit mine, the truck transportation cost accounts for more than 50% of the total production cost, and the truck fuel consumption cost is the main component of the truck transportation cost, and the fuel consumption cost accounts for 50~70% of the transportation cost [5]. With the continuous advancement of intelligent construction of open-pit mines, refined control of truck fuel consumption has attracted more and more attention from the industry. In recent years, with

the continuous popularization of sensor technology, positioning technology, and Internet of Things technology, it has brought new opportunities and challenges for refined control of the fuel consumption of trucks used in open-pit mines.

With the increasingly tight global energy supply, optimizing the performance of truck transportation and improving the technical level of the driver's operation so as to reduce the energy consumption of trucks and improve the economic level of production have become the focus of the industry. In terms of truck performance optimization, Martyushev, N.V. [6] established a mathematical simulation model of an electric vehicle traction battery and studied the dynamic charge and discharge modes of a heavy-duty electric vehicle traction battery under urban cycle conditions and driving conditions outside the city. Wang G [7] studied the suspension structure characteristics of mining trucks under different operating conditions and determined the damping characteristics of the suspension system during loading. Wenying Li [8] conducted an in-depth analysis of the truck structure design process, performance index setting, performance development process, etc. Hongliang Li [9] established an energy management strategy based on dynamic programming to study the fuel economy of hybrid mining trucks. Jorge Hurel [10] modeled the McPherson suspension for a quarter of the vehicle model and optimized the transient response relationship of truck spring acceleration. In terms of the research on the driving behavior of truck drivers, Ningli Wu [11] analyzed and studied the driving style of conventional highway transport trucks and proposed a truck fuel consumption prediction model. Jiangsu Zhu [12] established a driving behavior model of truck drivers based on multiple regression analysis and obtained significant characteristic parameters that are strongly related to truck fuel consumption. Zarkadoula M [13] conducted a pilot program of ecological driving, and the results showed that when drivers adopted ecological driving strategies, fuel consumption could be saved by 4.35%. Kropiwnicki J [14] determined the direct dynamic response relationship between truck travel time and vehicle energy consumption.

However, in the production process of open-pit mines, the trucking link is relatively extensive, especially in terms of drivers' driving behavior. According to incomplete statistics, the fuel consumption of good and bad drivers during transportation can vary by up to 15% [15]. With the continuous extension of the stope to the deep, the transportation environment becomes more and more complex, and the poor driving behavior of drivers directly or indirectly leads to the gradual reduction of single bucket-truck discontinuous mining system advantages and the gradual increase of production costs. In addition, unsafe behaviors that drivers may have in the process of driving also lead to the enhancement of safety risks in the transportation link, which restricts the long-term development goal of an open-pit mine.

In view of this, under the premise of ensuring transportation safety in open-pit mines, in order to minimize transportation production costs and optimize driving behaviors of mining truck drivers, a comprehensive evaluation method of driving behaviors of mining truck drivers, which takes both safe driving and transportation costs as indicators as factor variables, is proposed. In the process of research, firstly, a scientific and reasonable data collection scheme is developed by taking some operation parameters of mining trucks as the collection object. Secondly, the RKNN (the KNN algorithm is an improved algorithm) algorithm of time series prediction and the wavelet analysis method are used as data analysis methods, and the original data is processed scientifically. Then, through the construction of the principal component analysis theory model, the key index system representing safe driving behavior and transportation cost is established to realize the automatic evaluation and scientific assessment of the driving behavior of mining truck drivers, and the evaluation system of "standard driving", "prudent driving", and "aggressive driving" of mining truck drivers is proposed and formulated. Finally, the fine management of production costs in the transportation link of an open-pit mine is promoted.

## 2. Methods and Steps

### 2.1. Engineering Background

The single bucket-truck discontinuous mining system has become the mainstream mining process in Chinese open-pit mines due to its many advantages, such as flexibility, strong adaptability, and large production capacity [16,17]. However, with the gradual extension of the stope to the deep, the geological conditions become more complex, resulting in an increasingly harsh environment for truck transportation and a continuous increase in transportation costs, thus gradually reducing the advantages of the single bucket-truck discontinuous mining system [18]. According to incomplete statistics at the open-pit mine site, different driving behaviors of truck drivers can result in fuel consumption differences of up to 15% while driving the same truck under the same road conditions. Considering that optimizing truck transportation costs can play a decisive role in reducing the overall production cost of an open-pit mine (taking the production cost ratio of an open-pit mine as an example, each cost situation is shown in Figure 1). Therefore, this paper takes an open-pit mine in China as the research object and, based on the collected mining truck running state data, studies and analyzes the dynamic variation rule between truck drivers' driving behavior and truck running fuel consumption. Then, combined with the key characteristic indexes of truck drivers' driving behavior, the optimal driving behavior of mining truck drivers under different working conditions and different transportation conditions is determined, which provides a scientific basis for reducing the transportation cost of open-pit mines and improving the transportation safety of open-pit mines.

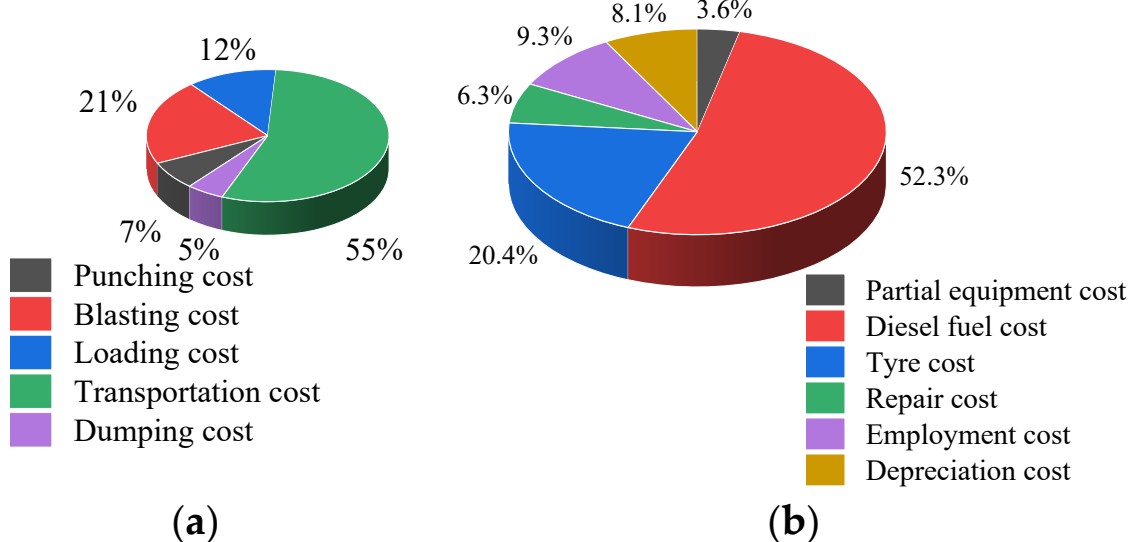

**Figure 1.** Schematic diagram of the production cost ratio of an open-pit mine. (**a**) Open-pit mine production links. (**b**) Open-pit mine transportation link.

### 2.2. Data Collection Method

2.2.1. Analysis of the Whole Process of Mining Truck Transportation

In the link of mining truck transportation in an open-pit mine, the purpose of the mining truck is mainly to transport the loose materials covered by the upper part of the ore body and loose ore body to the designated area so as to collect the ore body and process and utilize the ore body later. According to the analysis of the link between mining truck transportation and open-pit mines, the whole process of mining truck transportation can be divided into the following parts (taking mining truck transportation of loose rocks as an example, as shown in Figure 2).

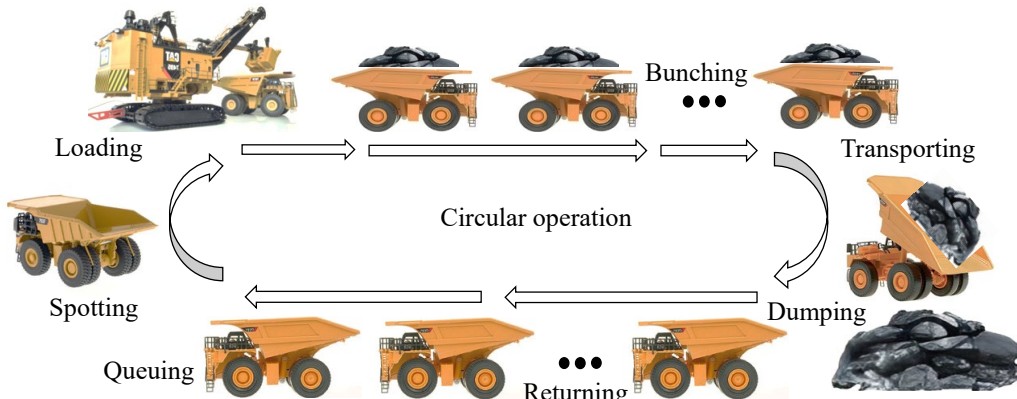

**Figure 2.** A schematic diagram of the whole process of mining truck transportation.

(1) Truck loading stage: the open-pit mine uses electric shovels or other excavators to load loose rocks into mining trucks;

(2) Truck bunching stage: a certain number of mining trucks are marshalled in an open-pit mine to serve different loading equipment;

(3) Heavy-duty transporting stage: After the loose rocks are transported to the designated area by mining trucks, the loose rocks are unloaded;

(4) No-load returning stage: After unloading all loose rocks, mining trucks start to return along the transport route;

(5) Truck queuing stage: mining trucks are transported to the loading area and queued up for loading equipment;

(6) Truck spotting stage: the mining truck enters the loading area and adjusts its attitude according to the spatial position of the loading equipment.

2.2.2. Design of the Data Acquisition Scheme

In order to accurately grasp the running state of mining trucks, the dynamic change law between the driving behavior of truck drivers and the fuel consumption of trucks was studied and determined. Data collected in this paper include spatial position data (error up to sub-meter level), fuel consumption data (error less than 0.5%, sampling frequency 1 Hz, accuracy up to mL level), vehicle body vibration data, driver pedal (accelerator, brake) opening data (sampling frequency up to 10 Hz), environmental meteorological data, carriage loading information, etc. The equipment layout scheme is shown in Figure 3. Where, 1 # represents the pedal; 2 # represents the top of the cockpit; 3 # represents the interior of the cockpit; 4 # represents the vehicle body skeleton; 5 # represents the hydraulic lifting device; 6 # represents the lower part of the carriage.

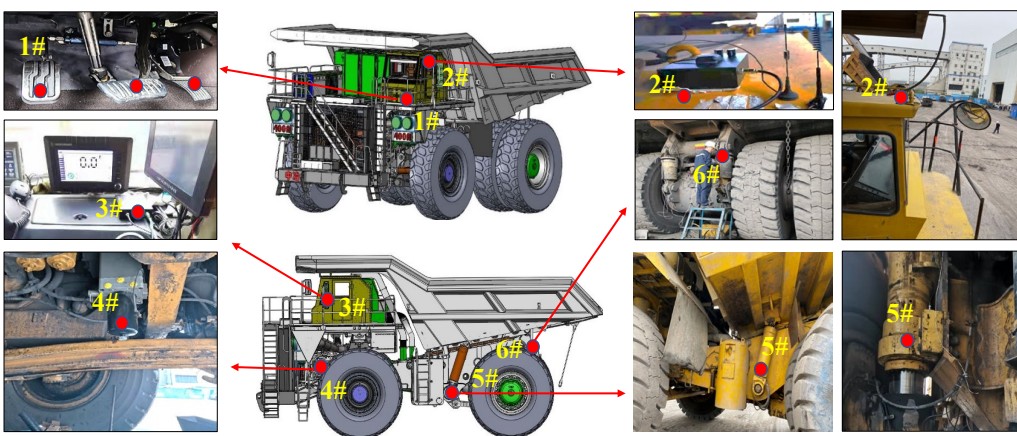

**Figure 3.** Schematic diagram of the data acquisition scheme.



Among them, position 1 mainly collects the angle data of the driver's pedal (accelerator and brake), and the selected equipment is the inclination sensor. Position 2 position mainly collects environmental meteorological data (temperature, humidity, and wind speed) and spatial location data. The selected equipment is an environmental meteorological sensor and a positioning sensor. Position 3 is a data analysis and storage module that mainly analyzes, processes, and stores various data indicators. Position 4 mainly collects fuel consumption data; the selected equipment is a fuel consumption sensor. Positions 5 and 6 mainly collect the vibration data of the car body and the state information of the carriage. The selected equipment is a vibration sensor.

### 2.3. Data Sample Processing Method

In the process of data collection, considering the complexity of the open-pit mine stope transportation environment, weak signals in local areas, and the large size of the mining truck leading to large seismic activity during the driving process, the massive data indexes collected by sensors may contain noise data, missing data, and redundant data [19,20]. Therefore, in order to minimize the deviation of data anomalies resulting in subsequent data analysis, this paper first needs to analyze and preprocess abnormal data before data analysis and data mining.

#### 2.3.1. Data Missing Processing Method Based on the RKNN Algorithm

Affected by multiple factors such as the complexity of the transportation environment, signal weakness, and strong seismic activity of the vehicle body, it is easy to cause the loss of multiple variable values of truck running state data in the process of data acquisition by multiple sensors; that is, at least two or more indicators in multiple groups of data are missing. The data missing in the multivariable missing mode is shown in matrix *X*.

$$X = \begin{bmatrix} x_{11} & x_{12} & * & \cdots & x_{1n-1} & x_{1n} \\ x_{21} & * & x_{23} & \cdots & * & * \\ * & x_{32} & * & \cdots & x_{3n-1} & x_{3n} \\ \cdots & \cdots & \cdots & \cdots & \cdots & \cdots \\ * & x_{m2} & x_{m3} & \cdots & * & x_{mn} \end{bmatrix} \tag{1}$$

where * represents the missing item of the variable; am n represents the data variable index; n represents the variable type; and m represents the sample size of the variable.

In a comprehensive consideration of the causes of missing data in mining truck running state prediction data, combined with the types of missing data information, this paper uses the RKNN algorithm to fill in the truck running state data, so as to increase the number of data samples and improve scientific support for subsequent data mining and analysis. The KNN (K-Nearest Neighbor) algorithm is one of the classical machine learning algorithms, and RKNN is a KNN [21,22] algorithm with an improved reduced association coefficient, which has many advantages such as simple principles and convenient operation. The main idea of the data filling algorithm is to regard the samples with missing data as the samples to be tested, and then select the K data closest to the missing sample data from the data set. Finally, the missing data is analyzed and processed according to the mean or mode of the K data, and the missing data index is determined. The specific calculation formula can be expressed as:

$$a_{ij} = \sqrt{\sum_{k=1}^{n} (x_{ik} - x_{ik})^2} \tag{2}$$

where $a_{ij}$ represents the Euclidean distance from sample *i* to sample *j* and $x_{ik}$ represents the data of sample *i* in the k-dimension.

The specific process of data filling by the RKNN algorithm can be divided into the following parts [23,24]:

Step 1: Firstly, the relevant data missing in the running state of the mining truck are pre-filled. If the missing data is continuous data, the average value is used to complete it. If the missing data is discrete data, the mode is used to complete it.

Step 2: Normalize all sample data to eliminate the impact of the data structure caused by dimensional problems.

Step 3: Formulas (3) and (4) were used to calculate the reduced correlation coefficient between the data to be filled and other samples.

$$RRC(x_{iq}, x_{jq}) = \frac{\rho Max_{\nabla q}\{|X_{iq} - \min_q|, |X_{iq} - \max_q|\}}{|X_{iq} - X_{jq}| + \rho Max_{\nabla q}\{|X_{iq} - \min_q|, |X_{iq} - \max_q|\}} \tag{3}$$

$$RRG(x_i, x_j) = \frac{1}{m}\sum_{q=1}^{m} RRC(x_{iq}, x_{jq}) \tag{4}$$

where $RRC(x_{iq}, x_{jq})$ represents the reduced correlation coefficient between sample *i* and sample *j* under index *q*; $x_{iq}$ and $x_{jq}$ represent the value of index *q* in the *i*th sample and the value of index *q* in the *j* sample, respectively; and $RRC(x_i, x_j)$ represents the reduced correlation coefficient between sample *i* and sample *j*.

Step 4: Order the determined reduced correlation coefficient from large to small and select K data samples that have been filled as reference data;

Step 5: Determine the type of missing data. For discrete data, mode is used to fill in, while for continuous data, mean value is used to fill in. Specific expressions are shown in Formulas (5) and (6).

$$\hat{y}_{ik} = \sum_{k=1}^{k} w_{ik} x_{ik} \tag{5}$$

$$w_{ik} = RRG(x_i, x_k) / \sum_{k=1}^{k} RRG(x_i, x_k) \tag{6}$$

where $w_{ik}$ represents the weight of the *k* adjacent sample with filling sample *i*.

After analyzing and processing the sample data using the time series prediction method, the sample number has significantly increased. Finally, after the successful implementation of the data acquisition scheme, six kinds of data information related to truck operation, such as elevation, speed, driving distance, instantaneous diesel consumption, cumulative diesel consumption, and diesel consumption during driving, can be collected (the environment-related parameter data mainly plays an auxiliary role in analysis). Taking the single transportation process of an open-pit mining truck as an example, a total of 837 sets of single factor variable data were obtained through the data acquisition device, and a total of 5859 sets of sample data were obtained, among which 856 sets of data were missing in the 5859 sets of data, and the collection accuracy was only 85.3%. After processing by the RKNN algorithm, the data sample size can reach about 5460, and the accuracy is increased by 7.9% (due to the limitation of the research space, the specific operation process will not be described). The running status data of some mining trucks are shown in Table 1.

Among them, $X_1$ represents the elevation; $X_2$ represents the traveling speed, *m/s*; $X_3$ represents the distance traveled, *m*; $X_4$ represents the instantaneous diesel consumption, *L/s*; $X_5$ represents the cumulative diesel consumption, *L*; and $X_6$ represents the diesel consumption during driving, *L*.

### 2.3.2. Data Denoising Processing Method Based on the Wavelet Analysis Method

Based on the RKNN algorithm, this paper realizes the filling of missing data in the sample data and improves the number of samples. In order to improve the accuracy of sample data and reduce the deviation of subsequent data analysis caused by noisy data as far as possible, this paper needs to further carry out noise reduction analysis and preprocessing of noisy data. Considering data noise types and data noise reduction

methods comprehensively, this paper adopts the wavelet transform method to realize noise reduction processing of sample data on mining truck running state [25].

**Table 1.** Running state data for some mining trucks.

| Position | $X_1$ | $X_2$ | $X_3$ | $X_4$ | $X_5$ | $X_6$ | ⋯ |
|----------|-------|-------|-------|-------|-------|-------|---|
| 1 | 1222 | 3.95 | 7.9 | 0.068 | 78.23 | 15.62 | ⋯ |
| 2 | 1222 | 4.06 | 16 | 0.065 | 78.34 | 15.73 | ⋯ |
| 3 | 1222 | 3.81 | 23.6 | 0.070 | 78.44 | 15.74 | ⋯ |
| 4 | 1221 | 3.95 | 31.5 | 0.060 | 78.55 | 15.85 | ⋯ |
| 5 | 1221 | 3.87 | 39.2 | 0.057 | 78.53 | 15.96 | ⋯ |
| 6 | 1220 | 3.84 | 46.9 | 0.040 | 78.62 | 15.94 | ⋯ |
| 7 | 1220 | 3.73 | 54.3 | 0.052 | 78.76 | 16.25 | ⋯ |
| 8 | 1219 | 3.90 | 62.1 | 0.046 | 78.75 | 16.16 | ⋯ |
| 9 | 1219 | 4.00 | 70.1 | 0.053 | 78.86 | 16.25 | ⋯ |
| 10 | 1218 | 4.14 | 78.4 | 0.068 | 78.97 | 16.24 | ⋯ |
| 11 | 1219 | 4.17 | 86.7 | 0.052 | 78.95 | 16.33 | ⋯ |
| 12 | 1218 | 4.23 | 95.2 | 0.047 | 79.23 | 16.42 | ⋯ |
| ⋯ | ⋯ | ⋯ | ⋯ | ⋯ | ⋯ | ⋯ | ⋯ |

According to the characteristics of mining truck running state data signal and noise at different scales, wavelet analysis can be processed to realize weak signal detection and enhance the signal-to-noise ratio, respectively, so as to complete the separation of real signal and noise [26]. The wavelet transform can effectively describe signal characteristics, so the use of wavelet noise reduction technology can not only remove the noise to the maximum extent but also ensure that the signal, after noise reduction, maintains the original characteristics.

The basic idea of a wavelet transform is to transform the signal $f(t)$ by using the wavelet basis function $\psi(t)$ [27]. For any signal $f(t) \in L_2(R)$, its continuous wavelet transform can be expressed as:

$$W_f(a,b) = \frac{1}{\sqrt{a}} \int_{-\infty}^{+\infty} f(t)\psi\left(\frac{t-b}{a}\right) dt, (a, b \in R, a \neq 0) \tag{7}$$

Data reconstruction can be expressed as:

$$f(t) = \frac{1}{C_\psi} \int_{-\infty}^{+\infty} \int_{-\infty}^{+\infty} W_f(a,b)\psi\left(\frac{t-b}{a}\right) dadb \tag{8}$$

where $a$ represents the scale parameter of data transformation, $b$ represents the data transformation translation parameter, and $C_\psi$ represents the wavelet transform coefficient.

The processing process of mining truck running state data signal denoising based on the wavelet analysis method is as follows:

(1) Signal multi-scale decomposition

The N-layer wavelet decomposition of mining truck running state data is carried out by choosing a wavelet function. That is, $F_{N-M}$ and $g_j$ can be determined by known $f_N$, where $j = N - 1, N - 2, \cdots, N - M$.

$$f_N = g_{N-1} + g_{N-2} + \cdots + g_{N-M} + f_{N-M} \tag{9}$$

where $f_i \in V_j$, $g_j \in W_j$, $f_N$ are $f \in L^2(R)$. The decomposition formula can be expressed as:

$$\begin{cases} c_{j+1}(n) = \sum\limits_{k \in Z} h(k-2n)c_j(k) \\ d_{j+1}(n) = \sum\limits_{k \in Z} g(k-2n)d_j(k) \end{cases} \tag{10}$$

where $j$ represents the number of original signal decomposition layers; $n$ represents the number of original signal sampling points; and $h$ and $g$ represent the orthogonal filter bank.

(2) The selection of the wavelet threshold

The signal noise of truck running state data is unknown, so it needs to be estimated. Considering the GCV (Generalized Cross Validation) method only through the data input and data output to determine the threshold value, it has nothing to do with noise ability or the real data. Therefore, the GCV method is adopted in this paper to determine the noise reduction threshold. The GCV expression is as follows:

$$GCV(\delta) = \frac{\|W - W_\delta\|^2}{N(N_0/N)^2} \tag{11}$$

where $N$ represents the number of wavelet coefficients, $N_0$ represents the number of wavelet coefficients 0, $W$ represents the wavelet coefficient of the signal input polluted by noise, and $W_\delta$ represents the wavelet coefficient after threshold processing.

Then the optimal threshold can be expressed as:

$$T = \mathrm{argmin}GCV(\delta) \tag{12}$$

(3) Wavelet reconstruction process

Contrary to signal decomposition, wavelet reconstruction solves $f_N$ with known $g_j$ and $F_{N-M}$, and the relevant expression is as follows:

$$\begin{cases} c_{kj} = \sum\limits_{l}\left[ p_{k-2l}c_{l,j-1} + q_{k-2l}d_{l,j-1} \right] \\ c_j = \left\{ c_{kj} \right\} \in l^2 \end{cases} \tag{13}$$

where $p(z)$ and $q(z)$ represent the corresponding scale functions.

In order to improve the data accuracy and verify the reliability of the wavelet analysis method, this paper takes the measured mining truck running state data as an example for data noise reduction. Considering the wavelet transform basis function, the DB5 wavelet is orthogonal, tightly supported, an approximately symmetric wavelet, with a linear phase, good smoothness, simple calculation, and many other advantages. In this paper, the DB5 wavelet is used to reduce the noise of the mining truck running state data signal. Through multi-scale decomposition, wavelet threshold selection, and wavelet reconstruction of data signals, real data can be obtained to characterize the running state information of mining trucks. After wavelet analysis of 5260 sets of sample data processed by the RKNN algorithm, the data accuracy is expected to increase by 10.3% after comparison and analysis with the original data. Taking the result of some data noise reduction processing as an example, the results are shown in Figures 4 and 5.

### 2.4. Evaluation Method of Driving Behavior of Mining Truck Drivers

The evaluation of drivers' driving behavior is the most basic performance evaluation method, aiming at the safety, economy, and efficient production of open-pit mine transportation links. The key to the scientific and rationality of performance evaluation lies in the selection of the mining truck driving behavior evaluation method. With the deepening research on driver's driving behavior evaluation in the process of conventional highway transportation, a large number of scholars have carried out a lot of research and discussion on the basic theory and application practice of the driver's driving behavior evaluation method. However, considering that the transport road of an open-pit mine is in the process of constant dynamic change with the advance of stope and the transport road construction is formed by rolling sand, stone, loess, and other materials, the road quality may be damaged at any time due to the rolling of equipment, thus leading to the uncertainty regarding road location and quality of open-pit mines. Therefore, limited by the uncertainties of transportation roads, there are few studies investigating the driving behav-

ior of mining truck drivers in open-pit mines. In addition, in the process of evaluating the driving behavior of mining truck drivers in an open-pit mine, there are many long-existing problems that cannot be solved effectively, such as insufficient driving information of trucks, serious lack of driving operation information of drivers, a complex and changeable layout of the development transportation system, etc. Although technicians and truck drivers collect and record some relevant information in the production process, the accuracy and effectiveness of the information cannot be guaranteed due to the roughness and delay of the recording. At the same time, there are significant differences in the comprehensive quality and experience of different technical personnel, which makes it difficult for management decision-makers to obtain accurate and effective data information and make scientific and reasonable evaluations of the performance appraisals of mining truck drivers.

Therefore, by combining and analyzing previous studies, this paper proposes a method to evaluate mining truck drivers' driving behavior based on the principal component analysis method, so as to provide reliable support for the evaluation and evaluation of mining truck drivers' driving behavior and improve the automation and information level of production management in an open-pit mine.

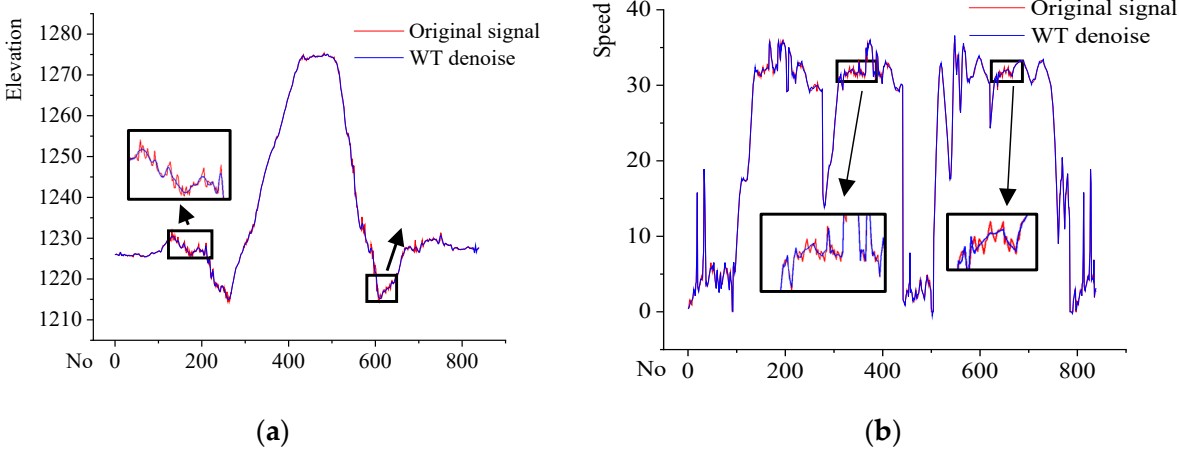

**Figure 4.** Part data before and after noise reduction of running speed and position of truck No. 1. (**a**) Partial running speed data of truck No. 1. (**b**) Partial operation location data of truck No. 1.

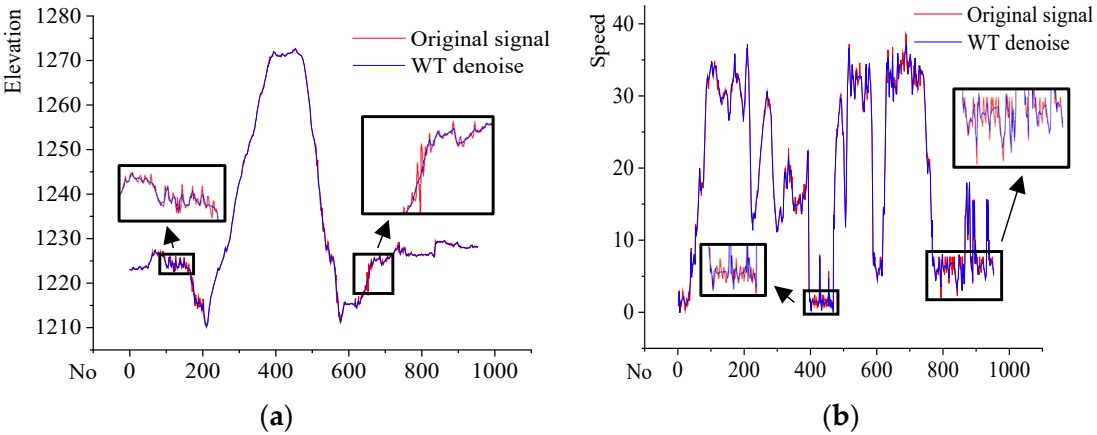

**Figure 5.** Part data before and after noise reduction of running speed and position of truck No. 2. (**a**). Partial running speed data of truck No. 2. (**b**). Partial operation location data of truck No. 2.

### 2.4.1. Principles of Principal Component Analysis

Based on mathematical theory, principal component analysis recombines and classifies a large number of relevant indicators (assuming a total of p indicators) to form a comprehensive index $F_m$ with limited numbers and no correlation to replace the original index [28,29].

The purpose of the extraction of comprehensive indicators is to not only ensure that the new indicators remain irrelevant to each other (information does not overlap), but also to make them reflect the information represented by the original variable *Xp* to the greatest extent. The specific operation process is as follows [30]:

First of all, it is assumed that there are n samples of data information related to mining truck drivers collected in the research, and each sample involves *p* index variables to form an $n \times p$ order data matrix:

$$X = \begin{bmatrix} x_{11} & x_{12} & \cdots & x_{1p} \\ x_{21} & x_{22} & \cdots & x_{2p} \\ \vdots & \vdots & \cdots & \vdots \\ x_{n1} & x_{n2} & \cdots & x_{np} \end{bmatrix} \tag{14}$$

Secondly, $F_1$ is set as the first principal component index formed by linear combination in the original variable, namely $F_1 = a_{11}x_1 + va_{12}x_2 + \cdots + a_{1p}x_p$. Then, the information extracted by principal component $F_1$ is effectively measured by variance. If the variance $Var(F_1)$ is positively correlated with the information contained in it, the larger the variance, the more information, and the more favorable it will be for the research and analysis of truck drivers' driving behavior.

Finally, in the process of analysis, if the original *p* indicates incomplete information for the first principal component, then there is a need to further select the second principal component index $F_2$; and the index of $F_1$ and $F_2$ are two indicators that are independent of each other and have no relevance, namely covariance Cor $(F_1, F_2) = 0$. On this basis, $F_1$, $F_2, \cdots, F_m$ are gradually constructed as the main components of the original variable index $x_1, x_2, \cdots, x_p$.

$$\begin{cases} F_1 = a_{11}x_1 + a_{12}x_2 + \ldots a_{1p}x_p \\ F_2 = a_{21}x_1 + a_{22}x_2 + \ldots a_{2p}x_p \\ \cdots \\ F_m = a_{m1}x_1 + a_{m2}x_2 + \ldots a_{mp}x_p \end{cases} \tag{15}$$

2.4.2. Calculation Procedure for Principal Component Analysis

Principal component analysis is based on the idea of dimensionality reduction, which gradually converts diversified indicators into a small number of comprehensive indicators, namely corresponding principal components. Each principal component can accurately and intuitively reflect most of the information on the condition that the information is not repeated [31,32]. This method can greatly eliminate the interference of human factors and ensure the objectivity, scientific accuracy, and effectiveness of the evaluation process and results. It is suitable for the evaluation system with closely related and systematic evaluation indicators. The specific analysis process is as follows [33]:

(1) Step 1: original data standardization processing

It is assumed that there are m index variables that can characterize the driving behavior of mining truck drivers, namely $x_1, x_2, \cdots, x_m$. There are *n* evaluation objects in total, and the value of the *j* index of the *i*th evaluation object is $x_{ij}$. Will each acquire $x_{ij}$ into a standardized index $\widetilde{x}_{ij}$ relation can be expressed as:

$$\widetilde{x}_{ij} = \frac{x_{ij} - \overline{x}_j}{s_j}, (i = 1, 2, \cdots, n; j = 1, 2, \cdots, m) \tag{16}$$

Among them, $\overline{x}_j =, \frac{1}{n} \sum_{i=1}^{n} x_{ij}, s_j = \sqrt{\frac{1}{n-1} \sum_{i=1}^{n} (x_{ij} - \overline{x}_j)^2}, (j = 1, 2, \cdots, m)$, and $\overline{x}_j, s_j$, indicate for the first *j*, the index, the sample mean, and the standard deviation, respectively.

(2) Step 2: Obtain the correlation coefficient matrix $R$

After standardizing the index data representing the driving behavior of mining truck drivers, it is necessary to further calculate and obtain the correlation coefficient matrix R, whose relationship can be expressed as:

$$R = (r_{ij})_{m \times m} \tag{17}$$

Among them, $r_{ij} = \frac{\sum\limits_{k=1}^{n} \tilde{x}_{ki} \times \tilde{x}_{kj}}{n-1}$, $(i, j = 1, 2, \cdots, m)$, $r_{ii} = 1$, $r_{ij} = r_{ji}$, and $r_{ij}$ are the correlation coefficients between the $i$th index and the $j$th index.

(3) Step 3: Determine the eigenvalues and eigenvectors

Using the step 2 calculation of the correlation coefficient matrix, $R$ eigenvalues from large to small are $\lambda_1 \geq \lambda_2 \geq \lambda_3 \geq \cdots \geq \lambda_m \geq 0$, and the corresponding eigenvector $u_1, u_2, \cdots$, $u_m$, where $u_j = (u_{1j}, u_{2j}, \cdots, u_{nj})^T$, composed of m new index variables can be expressed as:

$$\begin{cases} y_1 = u_{11}\tilde{x}_1 + u_{21}\tilde{x}_2 + \cdots u_{n1}\tilde{x}_n \\ y_2 = u_{12}\tilde{x}_1 + u_{22}\tilde{x}_2 + \cdots u_{n2}\tilde{x}_n \\ \cdots\cdots\cdots \\ y_m = u_{1m}\tilde{x}_1 + u_{2m}\tilde{x}_2 + \cdots u_{nm}\tilde{x}_n \end{cases} \tag{18}$$

where $y_1$ represents the first principal component, $y_2$ represents the second principal component, and $y_m$ represents the principal component.

(4) Step 4: Select $p$ $(p \leq m)$ principal components to calculate the comprehensive evaluation value

Finally, the corresponding information contribution rate and cumulative contribution rate are calculated according to the eigenvalue determined in Step 3. Then, through the information contribution rate and cumulative contribution rate of each principal component, the comprehensive evaluation value corresponding to the driving behavior samples of each group of mining truck drivers can be calculated. The specific calculation process is as follows:

Then, the principal component $y_i$ information contribution rate, which can represent the driving behavior index of mining truck drivers, can be calculated as follows:

$$b_j = \lambda_j / \sum_{k=1}^{m} \lambda_k, (j = 1, 2, \cdots, m) \tag{19}$$

The cumulative contribution rate of each principal component $y_1$, $y_2$, $\cdots$, $y_p$ can be calculated as follows:

$$\alpha_p = \sum_{k=1}^{p} \lambda_k / \sum_{k=1}^{m} \lambda_k \tag{20}$$

Among them, when $\alpha_p$ is close to 1 ($\alpha_p = 0.85, 0.90, 0.95$), the first $p$ index variables $y_1$, $y_2$, $\cdots$, $y_p$ are selected as p principal components to replace the original $m$ index variables, so as to conduct comprehensive analysis and calculation of p principal components.

In view of this, the formula for a comprehensive evaluation of the driving behavior samples of each group of mining truck drivers is as follows:

$$Z = \sum_{j=1}^{p} b_j y_j, (j = 1, 2, \cdots, m) \tag{21}$$

### 2.4.3. Establishment of a Driver Behavior Index Evaluation System

The accuracy and perfection of the index evaluation system play a decisive role in the subsequent evaluation of the driving behavior of mining truck drivers. Up to now, a perfect driver behavior index evaluation system has been developed in the field of conventional road transportation. Therefore, based on the improved index evaluation system and the

production status of the open-pit mining transportation link, this paper studies and puts forward the driving behavior index system of mining truck drivers, which takes into account both safe driving behavior and the production cost of the transportation link. Then, according to each index attribute, it is divided into two categories, which are related to the parameters related to safe driving behavior and the parameters related to the production cost of transportation.

(1) Analysis of indicators related to safe driving behavior

The index related to the acceleration or deceleration process can be used to describe the speed change of a mining truck and can judge whether the driver of the truck has extreme behavior such as rapid acceleration, rapid deceleration, and sudden braking. At the same time, it can judge the abnormal operation behavior of mining truck drivers under abnormal working conditions. In this paper, six indicators such as average speed driving ratio, overspeed ratio, rapid acceleration or deceleration ratio, and sudden braking ratio of mining trucks are selected as objects to carry out research and analysis, and the relevant expressions are shown in Table 2.

**Table 2.** Driving behavior indicators related to safe driving behavior.

| No | Name | Expression |
|----|------|------------|
| 1 | Maximum acceleration | $a_t(\max) = \{a_1, a_2, \cdots, a_n\}_{\max}$ |
| 2 | Minimum acceleration | $a_t(\min) = \{a_1, a_2, \cdots, a_n\}_{\min}$ |
| 3 | Mean acceleration | $\overline{a_t} = \frac{1}{n} \sum\limits_{t=1}^{n} a_t$ |
| 4 | Acceleration standard deviation | $\sigma_a = \sqrt{\frac{1}{n} \sum\limits_{t=1}^{n} (a_t - \overline{a_t})^2}$ |
| 5 | Overspeed ratio | $Speed = t_s / T$ |
| 6 | Sharp acceleration and deceleration ratio | $Rapid = t_r / T$ |
| 7 | Screeching ratio | $Quick = t_q / T$ |

(2) Analysis of indicators related to production costs in transportation links

As a key indicator to evaluate drivers' driving behavior, driving speed determines the fuel consumption of trucks at different time intervals. Therefore, speed-related indicators are taken as some key parameters to measure driving behavior. Then, according to the characteristics of truck operation in the transport link of an open-pit mine, this paper finally selects six indicators, such as maximum speed, minimum speed, average speed, and standard difference of speed, as its key parameters, and their relevant expressions are shown in Table 3:

**Table 3.** Driving behavior indicators related to speed.

| No | Name | Expression |
|----|------|------------|
| 1 | Maximum driving speed | $V_{\max} = \{v_1, v_2, \cdots, v_m\}_{\max}$ |
| 2 | Minimum travel speed | $V_{\min} = \{v_1, v_2, \cdots, v_m\}_{\min}$ |
| 3 | Average travel speed | $\overline{v} = \frac{1}{n} \sum\limits_{i=1}^{n} v_i$ |
| 4 | Velocity standard deviation | $\sigma_v = \sqrt{\frac{1}{n} \sum\limits_{i=1}^{n} (v_i - \overline{v})^2}$ |
| 5 | Uniform ratio | $Con = t_c / T$ |
| 6 | Single cycle oil consumption | $Q = \sum\limits_{i=1}^{t} q_i$ |

## 3. Results

In this paper, mining trucks are taken as the research object, and the data collection process is the heavy-duty transportation stage of mining trucks. Considering the numerous

factors that affect drivers' driving behaviors, in order to reflect the effectiveness of data comparison results, the idea of control variables is adopted to reduce the dimension of the data as much as possible. Therefore, in the process of data collection, drivers with similar ages and similar driving experience, working years, working time, and other indicators are selected. In terms of transport environment, routes with similar working conditions, such as transport road quality, transport road length, transport road slope, and so on, are selected. In terms of equipment type, mining trucks with similar conditions such as body load, service life, body working conditions, and vehicle performance are selected. Finally, after data pre-screening, only six groups of valuable data were obtained on the same road at the same time. After data analysis and pre-processing, data information on the heavy load transportation stage of mining trucks was obtained, and some of the data are shown in Table 4.

**Table 4.** Data on the heavy-duty transport stage of some mining trucks.

| No | $X_1$ | $X_2$ | $X_3$ | $X_4$ | $X_5$ | $X_6$ | $X_7$ | $X_8$ | $X_9$ | $X_{10}$ | $X_{11}$ | $X_{12}$ | $X_{13}$ |
|---|---|---|---|---|---|---|---|---|---|---|---|---|---|
| 1 | 9.56 | 7.56 | 6.45 | 1.47 | 0.237 | 6.722 | 0.50 | 0.89 | 0.20 | 0.07 | 1.4 | 0.20 | 0.50 |
| 2 | 10.17 | 7.14 | 7.90 | 1.28 | 0.239 | 8.921 | 0.80 | 0.88 | 0.20 | 0.03 | 2.6 | 0.26 | 0.80 |
| 3 | 9.61 | 7.08 | 7.95 | 1.61 | 0.241 | 9.537 | 0.50 | 0.9 | 0.10 | 0.02 | 2.8 | 0.28 | 0.50 |
| 4 | 9.67 | 7.42 | 6.95 | 2.17 | 0.245 | 8.211 | 0.40 | 0.92 | 0.20 | 0.02 | 3.2 | 0.28 | 0.40 |
| 5 | 9.83 | 7.53 | 6.5 | 1.56 | 0.268 | 8.105 | 0.20 | 0.93 | 0.10 | 0.03 | 2.8 | 0.32 | 0.20 |
| 6 | 9.67 | 7.03 | 6.58 | 2.33 | 0.118 | 8.951 | 0.30 | 0.91 | 0.20 | 0.06 | 1.8 | 0.20 | 0.30 |

Where $X_1$ represents the maximum traveling speed, $m/s$; $X_2$ represents the average travel speed, $m/s$; $X_3$ represents the standard deviation of the velocity; $X_4$ represents the minimum travel speed, $m$/s; $X_5$ represents the overspeed ratio; $X_6$ represents fuel consumption, $L$; $X_7$ represents the minimum acceleration, $m/s^2$; $X_8$ represents the ratio of uniform speed; $X_9$ represents the ratio of rapid acceleration; $X_{10}$ represents the proportion of sudden braking; $X_{11}$ represents the maximum acceleration, $m/s^2$; $X_{12}$ represents the standard deviation of acceleration; and $X_{13}$ represents the mean acceleration, $m/s^2$ (the contents of each related variable are the same below).

After processing the data information in Table 5 according to Formulas (16) and (17), the correlation matrix of the data in the heavy-duty transport stage of the mining truck in the transport link of the open-pit mine is obtained, as shown in Table 5 below.

**Table 5.** Data correlation matrix of the heavy-duty transport stage of a mining truck.

| Correlation Matrix | | | | | | | | | | | | | |
|---|---|---|---|---|---|---|---|---|---|---|---|---|---|
| Index | $X_1$ | $X_2$ | $X_3$ | $X_4$ | $X_5$ | $X_6$ | $X_7$ | $X_8$ | $X_9$ | $X_{10}$ | $X_{11}$ | $X_{12}$ | $X_{13}$ |
| $X_1$ | 1 | −0.207 | 0.445 | −0.477 | 0.202 | 0.301 | 0.538 | −0.305 | 0.096 | −0.336 | 0.310 | 0.320 | 0.538 |
| $X_2$ | −0.207 | 1 | −0.602 | −0.244 | 0.613 | −0.864 | −0.313 | 0.334 | −0.046 | 0.138 | −0.031 | 0.221 | −0.313 |
| $X_3$ | 0.445 | −0.602 | 1 | −0.391 | 0.235 | 0.701 | 0.728 | −0.527 | −0.189 | −0.623 | 0.465 | 0.283 | 0.728 |
| $X_4$ | −0.477 | −0.244 | −0.391 | 1 | −0.669 | 0.177 | −0.568 | 0.537 | 0.284 | 0.097 | 0.025 | −0.247 | −0.568 |
| $X_5$ | 0.202 | 0.613 | 0.235 | −0.669 | 1 | −0.267 | 0.191 | 0.046 | −0.432 | −0.538 | 0.516 | 0.700 | 0.191 |
| $X_6$ | 0.301 | −0.864 | 0.701 | 0.177 | −0.267 | 1 | 0.153 | −0.011 | −0.326 | −0.577 | 0.486 | 0.293 | 0.153 |
| $X_7$ | 0.538 | −0.313 | 0.728 | −0.568 | 0.191 | 0.153 | 1 | −0.902 | 0.374 | −0.113 | −0.014 | −0.181 | 1 |
| $X_8$ | −0.305 | 0.334 | −0.527 | 0.537 | 0.046 | −0.011 | −0.902 | 1 | −0.414 | −0.275 | 0.421 | 0.512 | −0.902 |
| $X_9$ | 0.096 | −0.046 | −0.189 | 0.284 | −0.432 | −0.326 | 0.374 | −0.414 | 1 | 0.483 | −0.414 | −0.699 | 0.374 |
| $X_{10}$ | −0.336 | 0.138 | −0.623 | 0.097 | −0.538 | −0.577 | −0.113 | −0.275 | 0.483 | 1 | −0.978 | −0.864 | −0.113 |
| $X_{11}$ | 0.310 | −0.031 | 0.465 | 0.025 | 0.516 | 0.486 | −0.014 | 0.421 | −0.414 | −0.978 | 1 | 0.878 | −0.014 |
| $X_{12}$ | 0.320 | 0.221 | 0.283 | −0.247 | 0.700 | 0.293 | −0.181 | 0.512 | −0.699 | −0.864 | 0.878 | 1 | −0.181 |
| $X_{13}$ | 0.538 | −0.313 | 0.728 | −0.568 | 0.191 | 0.153 | 1 | −0.902 | 0.374 | −0.113 | −0.014 | −0.181 | 1 |

In order to accurately evaluate the comprehensive driving state of mining truck drivers in the heavy-duty transportation stage of an open-pit mine, principal component analysis of sample data is needed after correlation analysis of data indexes. According to the

principle of principal component analysis, the principle of cumulative contribution rate $\alpha_p \geq 85\%$ was strictly followed, and the data were processed using Formulas (18) to (20). The cumulative contribution rate of the first three principal components was calculated as high as 87.820% (>85%). Therefore, the first three principal components basically represent 89% of the information content of the 13 driving behavior characteristics. The specific principal component characteristic value and cumulative contribution rate are shown in Table 6.

**Table 6.** Principal component characteristic value and cumulative contribution rate.

| Composition | Eigenvalue | Contribution Rate | Cumulative Contribution% | Total | Variance % | Accumulation% |
|---|---|---|---|---|---|---|
| 1 | 4.7010 | 36.1615 | 36.162 | 4.701 | 36.162 | 36.162 |
| 2 | 4.1361 | 31.8165 | 67.978 | 4.136 | 31.816 | 67.978 |
| 3 | 2.5795 | 19.8421 | 87.820 | 2.579 | 19.842 | 87.820 |
| 4 | 0.9332 | 7.1781 | 94.998 | | | |
| 5 | 0.6502 | 5.0018 | 100.000 | | | |
| 6 | 0.0000 | 0.0000 | 100.000 | | | |
| 7 | 0.0000 | 0.0000 | 100.000 | | | |
| 8 | 0.0000 | 0.0000 | 100.000 | | | |
| 9 | 0.0000 | 0.0000 | 100.000 | | | |
| 10 | (0.0000) | (0.0000) | 100.000 | | | |
| 11 | (0.0000) | (0.0000) | 100.000 | | | |
| 12 | (0.0000) | (0.0000) | 100.000 | | | |
| 13 | (0.0000) | (0.0000) | 100.000 | | | |

According to the analysis in Table 6:

(1) The characteristic value of principal component 1 is 4.7010, and the contribution rate is 36.162%. The driving behavior characteristics of drivers with higher loads are maximum driving speed, minimum acceleration, average acceleration, standard deviation of speed, and uniform acceleration ratio column, which are all related to the driver's standard driving. Therefore, principal component 1 can be named "standard driving factor".

(2) The characteristic value of principal component 2 is 4.1361, and the contribution rate is 31.817%. The driving behavior characteristics of drivers with higher loads are minimum driving speed, uniform speed ratio, maximum acceleration, and fuel consumption, which are all related to prudent driving. Principal component 2 can be named "prudent driving factor".

(3) The characteristic value of principal component 3 is 2.5795, and the contribution rate is 19.872%. The driving behavior characteristics of drivers with higher loads are average driving speed, overspeed ratio, sudden braking ratio, and rapid acceleration ratio, which reflect the aggressive driving information of drivers. Principal component 3 can be named "aggressive driving factor".

At the same time, based on the principal component matrix of the driving behavior sample data of mining truck drivers, the strength correlation analysis of each index of the three principal components is carried out so as to make a comprehensive evaluation of the driving behavior of mining truck drivers. The correlation analysis results are shown in Table 7.

Finally, according to Formula (21), this paper builds a comprehensive evaluation model for the heavy-duty transport stage of mining trucks. Then, according to the proportional weight of the eigenvalue of each principal component in the total eigenvalue of the extracted principal component, combined with the corresponding eigenvector value of each principal component, the final comprehensive evaluation value of the principal component factor can be obtained. Then the scientific evaluation of the driving behavior of mining truck drivers is realized. The final comprehensive evaluation values of the sample data is shown in Table 8.

**Table 7.** Correlation analysis of principal component indexes.

| Component Matrix | Composition | | |
|---|---|---|---|
| | **1** | **2** | **3** |
| $X_1$ | 0.672 | −0.137 | 0.079 |
| $X_2$ | −0.362 | 0.328 | 0.852 |
| $X_3$ | 0.926 | −0.145 | −0.245 |
| $X_4$ | −0.568 | 0.174 | −0.685 |
| $X_5$ | 0.478 | 0.437 | 0.741 |
| $X_6$ | 0.565 | 0.147 | −0.794 |
| $X_7$ | 0.713 | −0.663 | 0.161 |
| $X_8$ | −0.428 | 0.871 | −0.151 |
| $X_9$ | −0.239 | −0.711 | −0.022 |
| $X_{10}$ | −0.722 | −0.629 | 0.148 |
| $X_{11}$ | 0.596 | 0.706 | −0.134 |
| $X_{12}$ | 0.507 | 0.842 | 0.148 |
| $X_{13}$ | 0.713 | −0.663 | 0.161 |

**Table 8.** Sample data comprehensive evaluation and analysis table.

| No | Principal Component Factor Score | | | Composite Score | Ranking |
|---|---|---|---|---|---|
| | $F_1$ | $F_2$ | $F_3$ | | |
| 1 | 4.534464 | 1.464869 | 5.968043 | 3.746 | 4 |
| 2 | 5.856108 | 1.742691 | 4.889465 | 4.147 | 1 |
| 3 | 5.465452 | 1.991325 | 4.200941 | 3.921 | 2 |
| 4 | 4.81742 | 1.946713 | 4.540517 | 3.714 | 5 |
| 5 | 4.934976 | 1.699483 | 5.357386 | 3.858 | 3 |
| 6 | 4.633526 | 1.533592 | 3.748448 | 3.310 | 6 |

## 4. Discussion and Analysis

In order to develop a more scientific and reasonable driver behavior evaluation system, the assistant open-pit mine achieves fine control of mining truck oil consumption and alleviates the increasingly tight trend of global energy supply. This paper presents a comprehensive evaluation method of the driving behavior of mining truck drivers, which takes both safe driving and transportation costs as factors. However, considering that the transportation link of an open-pit mine is a complex and large system, the transportation environment is complex and changeable, there are many factors affecting the consumption of truck oil, and there is a certain gap in the driving ability of drivers. Therefore, in the process of research, there remains room for improvement, namely the following:

(1) In terms of data collection, although the implementation of the data collection scheme has already possessed the ability to collect and transmit the operation data of mining trucks, taking into account that the sensor, as a precision collection device, has strict requirements on environmental conditions, open-pit mines are generally located in areas with complex environments, so some data may have errors in the process of data collection.

(2) In terms of data analysis, this paper builds a time series prediction model and a wavelet analysis data analysis model. After data analysis and processing, the accuracy of mining truck operation data is improved by 7~12%. However, the relevant algorithms or models cannot be integrated with the operation of mining truck drivers during operation. In the data analysis and processing, noise reduction may be applied to the problem that the driver brakes sharply and causes the speed to decrease instantaneously. Thus, the algorithm model still has some limitations.

(3) In terms of data results, although the comprehensive evaluation method proposed in the study realizes the automated assessment of the driving behavior of mining truck drivers, the transportation cost is expected to be reduced by about 5% after the driver's driving operation is standardized. However, considering the driver's age, driving experience, driving years and working hours, and many other factors that have a crucial

impact on the driver's driving operation behavior, our analysis is restricted by the information confidentiality provisions of open-pit mine personnel. Therefore, only a small number of drivers with the same driving experience can be used as research objects in the research process, and the sample data is limited, which may lead to certain deviations in the assessment results.

## 5. Conclusions

Different from the previous evaluation studies of drivers' driving behavior in open-pit mines, which mainly took safe driving behavior index as a factor variable, this paper proposes a comprehensive evaluation method of mining truck drivers' driving behavior that takes both safe driving and transportation cost as factor variables, which laid a certain foundation for realizing fine control of mining truck oil consumption in open-pit mines and reducing the production cost of open-pit mines.

The research results show that: (1) The comprehensive evaluation system of driving behavior of mining truck drivers, which takes into account both safe driving and transportation costs, is obtained in this study. This allowed for the evaluation of the driving behavior of road transport drivers with fixed transport routes and good transport road quality and provides a new idea and approach for the research of driving behavior of mining truck drivers with dynamic transport routes and complex transport road conditions. (2) In view of the problems due to missing data, data redundancy, and data errors in the original data, a data missing processing and data noise reduction method based on time series prediction and wavelet analysis is proposed. After the original data processing, the data accuracy can be improved by approximately 7–12%. The noise reduction method provides a reference for data processing in other fields. (3) Taking the principal component analysis method as the starting point, the evaluation system of "standard driving", "prudent driving", and "aggressive driving" driving behavior of mining truck drivers is researched and formulated, and the automatic assessment of driving behavior of mining truck drivers is realized. After the operation behavior of drivers is standardized, the transportation cost is expected to be reduced by about 5%.

**Author Contributions:** Conceptualization, Z.Z.; Formal analysis, R.Z.; Investigation, J.S. All authors have read and agreed to the published version of the manuscript.

**Funding:** The work was supported by the basic scientific research service fee of central universities (funding number: 3142019007).

**Institutional Review Board Statement:** Not applicable.

**Informed Consent Statement:** Informed consent was obtained from all subjects involved in the study.

**Data Availability Statement:** The data used to support the findings of this study are available from the corresponding author upon request.

**Conflicts of Interest:** The authors declare no conflict of interest.

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
