# Peer review of "Research on the Comprehensive Evaluation Method of Driving Behavior of Mining Truck Drivers in an Open-Pit Mine"

_applsci, doi:10.3390/app132011597_

Round 1

Reviewer 1 Report

Comments for Authors

Dear Authors,

First of all, thanks for contributing with this interesting research about on comprehensive evaluation method of driving behaviour of mining truck drivers in open-pit mine.

After a careful assessment of the paper, I believe the reviewed manuscript addresses a pertinent research problem (i.e. method for driving behaviour evaluation) for being considered as publishable in Applied. Of course, some points need to be addressed beforehand.

In general, the structure of the paper is adequate; however, the presentation can be clearer and more concise. In this regard, some concerns, queries and suggestions raised during my review must be addressed, in order to optimize the manuscript contents and its suitability for the journal:

GENERAL OBSERVATION: It´s not really clear which is the main objective of this study: driving behaviour? In which areas: safety behaviour, risk perception? Or only fuel consumption, if is only fuel consumption. Please indicate this and be specific in what type of driving behaviour your research analyses, which I understand is just fuel consumption. Or your paper presents a comprehensive evaluation method of driving behaviour of mining truck drivers in open-pit mine?

In the paper it indicates that it has not been able to take into account the variable age of the drivers, it should also take into account the variable: level of training in driving,

You must also take into account for the results if the trucks are of the same age and are properly maintained, so that their operation is correct and driving is the same for everyone.

It is not clear what the result is; the evaluation method?

If it is the method, it should take into account other variables for the future such as:

·        Age

·        Driving experience

·        Driver training

·        Years worked in the mine

·        Work schedule and shifts

These variables are decisive in explaining the driver's behaviours in its three components.

Author Response

Dear expert:

     Thank you for your sincere guidance and review of the manuscript. Please refer to the attachment for the revised manuscript and revision instructions.

Reviewer 2 Report

The article is devoted to the study of a method for a comprehensive assessment of the driving behavior of drivers of mining dump trucks in a quarry. A method for collecting information about the state of mining dump trucks in a quarry is proposed, and the data collection results mainly correspond to the working state of mining dump trucks in a quarry. However, there are some comments regarding the work:

1. The Abstract section must be rewritten to reflect the relevance of the problem being solved and the scientific novelty of the solution obtained.

2. Keywords must be adjusted, highlighting special terms that characterize the study.

3. In the Introduction section, the relevance of the research being conducted should be indicated. At the end of the Introduction section, it is necessary to define the purpose of the scientific research and present the detailed structure of the article with a presentation of the problems to be solved in the following sections.

4. The list of cited sources should include more modern publications on the energy of movement of mining dump trucks, for example,

https://doi.org/10.3390/math11030536

https://doi.org/10.3390/IECMA2022-12904

5. When developing a data collection scheme, how were the parameters influencing the target fuel consumption function determined?

6. In the Discussion section, you should characterize the obtained models and the scientific results obtained, describe their advantages and disadvantages, and also give the limitations of the resulting model of the driver’s influence on fuel consumption.

7. A comparative analysis of the results obtained regarding fuel consumption by dump trucks with the results of other researchers based on information sources should be provided

8. Conclusions must be structured, highlighting the main scientific and practical results obtained, supporting them with numerical results.

Author Response

(The authors gave the same response as above.)

Reviewer 3 Report

The authors emphasize very well the existence of measurement errors and possible missing data during the data acquisition process. The use of reverse k nearest neighbors overcomes these issues and facilitates the obtaining of more accurate driving behavior evaluations. The evaluation considers various variables, such as dynamic routes and complex transport road conditions. 

The paper is well organized, but there are still some points that need to be clarified. Please see below my suggestions for improvement: 

1. Create the reference list according to the journal guidelines. 

2. Please add an extended version of the abbreviations used in the paper at their first appearance: RKNN (line 21, 91); GA, PSO, ANFID, LLNF, ADVISOR lines (55-67); TOPSIS (line 72); JTH (line 377). 

3. The data used for the evaluation of the cost ratio in Figure 1 are historical data? It is not clear the source of these data (lines 111-124). 

4. The data acquisition process for the experiment is very well described and simplifies the reproducibility of the experiment. 

5. There are some small typos (s with lowercase on line 103 and two dots on line 121). 

6. It is not clear whether the data used in the experiment were handled correctly. In Table 1 (including the explanations from lines 277-230) and Table 5 (including the explanations from lines 466-473) km/h for speed, m for distance, and m/s2 for acceleration are used. Please justify why for speed was not used m/s to be aligned with the use of the other measurement units as described by the international systems of units.  

7. Please check if the km/h values were correctly transformed into m/s during calculations to comply with the use of the same measurement units as for the other parameters. 

8. In the Discussion section, please explain how the results obtained are comparable with the state-of-the-art. 

Author Response

(The authors gave the same response as above.)

Round 2

Reviewer 2 Report

In general, the authors revised the article. The article may be published in my opinion. My comments have been taken into account.

Reviewer 3 Report

Dear authors,

Thank you for considering my suggestions in updating the paper. Considering that the data is currently correctly handled during experiments and the discussion and conclusion sections were updated accordingly, the paper can be accepted for publication.